# Trajectory of Humoral Responses to Two Doses of ChAdOx1 nCoV-19 Vaccination in Patients Receiving Maintenance Hemodialysis

Tsai-Chieh Ling,[a] Po-Lin Chen,[a,b,c] Nan-Yao Li,[a] Wen-Chien Ko,[a,c] Chien-Yao Sun,[a,d] Jo-Yen Chao,[a] Chi-Chang Shieh,[c,e,f] Ching-Fen Shen,[e,f] Jia-Ling Wu,[a,g] Teng-Ching Huang,[a] Chiao-Hsuan Chao,[h] Jen-Ren Wang,[h,i] ⓘ Yu-Tzu Chang[a]

aDepartment of Internal Medicine, National Cheng Kung University Hospital, College of Medicine, National Cheng Kung University, Tainan, Taiwan
bInfection Control Center, National Cheng Kung University Hospital, College of Medicine, National Cheng Kung University, Tainan, Taiwan
cDepartment of Medicine, College of Medicine, National Cheng Kung University, Tainan, Taiwan
dDepartment of Geriatrics and Gerontology, National Cheng Kung University Hospital, College of Medicine, National Cheng Kung University, Tainan, Taiwan
eDepartment of Pediatrics, National Cheng Kung University Hospital, College of Medicine, National Cheng Kung University, Tainan, Taiwan
fInstitute of Clinical Medicine, College of Medicine, National Cheng Kung University, Tainan, Taiwan
gDepartment of Public Health, College of Medicine, National Cheng Kung University, Tainan, Taiwan
hDepartment of Medical Laboratory Science and Biotechnology, College of Medicine, National Cheng Kung University, Tainan, Taiwan
iNational Institute of Infectious Diseases and Vaccinology, National Health Research Institutes, Tainan, Taiwan

**ABSTRACT** The ChAdOx1 nCoV-19 (AZD1222) vaccine is one of the most commonly delivered SARS-CoV-2 vaccines worldwide; however, few clinical studies have investigated its immunogenicity in dialysis patients. We prospectively enrolled 123 patients on maintenance hemodialysis at a medical center in Taiwan. All patients were infection-naive, had received two doses of the AZD1222 vaccine, and were monitored for 7 months. The primary outcomes were anti-SARS-CoV-2 receptor-binding domain (RBD) antibody concentrations before and after each dose and 5 months after the second dose and neutralization capacity against ancestral SARS-CoV-2, delta, and omicron variants. The anti-SARS-CoV-2 RBD antibody titers significantly increased with time following vaccination, with a peak at 1 month after the second dose (median titer, 498.8 U/mL; interquartile range, 162.5 to 1,050 U/mL), and a 4.7-fold decrease at 5 months. At 1 month after the second dose, 84.6, 83.7, and 1.6% of the participants had neutralizing antibodies against the ancestral virus, delta variant, and omicron variant, respectively, measured by a commercial surrogate neutralization assay. The geometric mean 50% pseudovirus neutralization titers for the ancestral virus, delta variant, and omicron variant were 639.1, 264.2, and 24.7, respectively. The anti-RBD antibody titers correlated well with neutralization capacity against the ancestral virus and delta variant. Transferrin saturation and C-reactive protein were associated with neutralization against the ancestral virus and delta variant. Although two doses of the AZD1222 vaccine initially elicited high anti-RBD antibody titers and neutralization against the ancestral virus and delta variant in hemodialysis patients, neutralizing antibodies against omicron variant were rarely detected, and the anti-RBD and neutralization antibodies waned over time. Additional/booster vaccinations are warranted in this population.

**IMPORTANCE** Patients with kidney failure have worse immune response following vaccination compared to general population, but few clinical studies have investigated immunogenicity of ChAdOx1 nCoV-19 (AZD1222) vaccination in hemodialysis patients. Here, we showed two doses of AZD1222 vaccines lead to high seroconversion rate of anti-SARS-CoV-2 receptor-binding domain (RBD) antibodies, and more than 80% patients acquired neutralizing antibodies against ancestral virus and delta variant. However, seldom did they obtain neutralizing antibodies against the omicron variant. The geometric mean 50% pseudovirus neutralization titer against the ancestral virus was 25.9-fold higher than that against the omicron variant. Also, there was a substantial decay in anti-RBD titers

**Ad Hoc Peer Reviewers** ⓘ Luciana Costa, Universidade Federal do Rio de Janeiro; Jianjun Chen, Wuhan Institute of Virology

Address correspondence to Yu-Tzu Chang, kangxiemperor@gmail.com.

The authors declare no conflict of interest.

with time. Our findings provided evidence supporting that more protective measures, including additional/booster vaccinations, is warranted in these patients during the current COVID-19 pandemic.

**KEYWORDS** COVID-19, ChAdOx1 nCoV-19 vaccine, anti-SARS-CoV-2 RBD antibody, neutralizing antibody, variants of concern, hemodialysis

The coronavirus disease 2019 (COVID-19) pandemic poses a major threat to patients with chronic kidney disease, as they are more prone to SARS-CoV-2 infection and the subsequent complications, including hospitalization, respiratory failure, and mortality. Patients treated with in-center hemodialysis (HD) are particularly vulnerable to COVID-19, since they cannot self-isolate during the dialysis procedure, and many of them are elderly with multiple comorbidities (1). An estimated 28 to 36% of kidney failure patients were infected during the first wave of the pandemic, which is 5 to 20 times higher than in the general population (2), and the mortality rate was as high as 20 to 30% in Europe (3) and even higher than 50% for some regions with lower income (4).

Several SARS-CoV-2 vaccines have been proven to reduce the risks of infection and severe disease in the general population; however, none of the efficacy trials included dialysis patients. Since responses to vaccines such as hepatitis B and influenza vaccines (5, 6) are weaker in patients undergoing dialysis, it is important to investigate whether authorized SARS-CoV-2 vaccines provide adequate protection for this population. Most studies investigating immunogenicity in dialysis patients have focused on humoral responses to structural proteins (i.e., spike protein [S] or receptor-binding domain [RBD]) following mRNA vaccines (7, 8). However, data regarding the ChAdOx1 nCoV-19 vaccine (AZD1222), one of the top three most commonly delivered SARS-CoV-2 vaccines worldwide, are scant. It is well known that antibody titers and protection decay over time after immunization (9, 10), and current vaccines have been shown to be less protective against several emerging SARS-CoV-2 variants (11, 12). The delta variant (B.1.617.2) was the predominant strain from mid-2021 until 2022, after which the omicron variant (B.1.1.529) became the leading strain worldwide (unpublished data). Neutralizing antibody levels are highly correlated with protection against infection (10, 12) and cellular immunity (13), and their correlation with breakthrough infection is stronger than that for anti-S or anti-RBD antibodies (14). In this study, we aimed to explore the humoral responses, including anti-SARS-CoV-2 RBD titers and viral neutralization tests against ancestral virus, the delta variant, and the omicron variant for up to 5 months after two doses of the AZD1222 vaccine in HD patients.

## RESULTS

**Baseline characteristics and adverse events following vaccination.** A total of 131 HD patients were initially enrolled, 123 of whom completed the scheduled blood sampling protocol (see Fig. S1 in the supplemental material). The baseline characteristics are shown in Table 1. There were significant differences in age and diabetes between the groups with inhibitions of ≥30% and <30% against the ancestral virus and the delta variant.

The most common adverse reaction to the vaccine following the first dose was fever, which developed in 26 (21.1%) patients, following by headache (4.9%), injection site pain (3.3%), fatigue (3.3%), and arthralgia/myalgia (3.3%) (see Fig. S2). Following the second dose, only two (1.6%) patients developed fever. No serious adverse events were noted during the study period.

**Secular changes in anti-SARS-CoV-2 RBD antibody levels over the 7-month follow-up period.** Anti-SARS-CoV-2 RBD antibody titers were measured at baseline (within 7 days before the first dose), V1M1 (1 month after the first dose), V1M2 (2 months after the first dose), V2M1 (1 month after the second dose), and V2M5 (5 months after the second dose). The anti-SARS-CoV-2 RBD antibody concentration significantly increased from the beginning of the study until V2M1, with median antibody titers of 0.30 U/mL (interquartile range [IQR] = 0.30 to 0.30), 6.22 U/mL (IQR = 0.87 to 25.45), 28.53 U/mL (IQR = 6.35 to 84.19) and 498.80 U/mL (IQR = 162.50 to 1,050.00) at baseline, V1M1, V1M2, and V2M1, respectively. The seroconversion rate at V2M1 was 98.37%. At V2M5, the median

**TABLE 1** Baseline characteristics of the enrolled vaccination groups[a]

| Variable | All patients (n = 123) | sVNT (ancestral type) | | | sVNT (delta variant) | | | sVNT (omicron variant) | | |
|---|---|---|---|---|---|---|---|---|---|---|
| | | Inhibition ≥30% (n = 104) | Inhibition <30% (n = 19) | P | Inhibition ≥30% (n = 103) | Inhibition <30% (n = 20) | P | Inhibition ≥30% (n = 2) | Inhibition <30% (n = 121) | P |
| Age, yr | 64.58 ± 13.39 | 63.10 ± 13.59 | 72.68 ± 8.72 | 0.0003 | 63.16 ± 13.68 | 71.90 ± 8.87 | 0.0008 | 75 ± 7.07 | 64.41 ± 13.41 | 0.2687 |
| Female gender | 58 (47.15) | 52 (50) | 6 (31.58) | 0.1391 | 53 (51.46) | 5 (25) | 0.0301 | 1 (50) | 64 (52.89) | 0.7976 |
| Dialysis vintage, yr | 5.41 ± 4.31 | 5.56 ± 4.56 | 4.58 ± 2.46 | 0.1829 | 5.70 ± 4.58 | 3.89 ± 1.96 | 0.0052 | 4.63 ± 2.65 | 5.42 ± 4.34 | |
| Diabetes mellitus | 65 (52.85) | 50 (48.08) | 15 (78.95) | 0.0132 | 50 (48.54) | 15 (75) | 0.0301 | 1 (50) | 64 (52.89) | |
| Hypertension | 118 (95.93) | 100 (96.15) | 18 (94.74) | 0.5744 | 98 (95.15) | 20 (100) | 0.5908 | 2 (100) | 116 (95.87) | |
| Heart failure | 32 (26.02) | 25 (24.04) | 7 (36.84) | 0.2623 | 26 (25.24) | 6 (30) | 0.6572 | 1 (50) | 31 (25.62) | |
| Cerebral vascular disease | 6 (4.88) | 4 (3.85) | 2 (10.53) | 0.2317 | 5 (4.85) | 1 (5) | 0.9779 | 0 (0) | 6 (4.96) | |
| COPD | 2 (1.63) | 2 (1.92) | 0 (0) | | 2 (1.94) | 0 (0) | | 0 (0) | 2 (1.65) | |
| Myocardial infarction | 5 (4.07) | 5 (4.81) | 0 (0) | | 5 (4.85) | 0 (0) | | 0 (0) | 5 (4.13) | |
| Cancer history | 19 (15.45) | 16 (15.38) | 3 (15.79) | 0.9642 | 17 (16.5) | 2 (10) | 0.7360 | 0 (0) | 19 (15.7) | |
| Disability | 15 (12.20) | 11 (10.58) | 4 (21.05) | 0.2473 | 13 (12.62) | 2 (10) | 0.7430 | 1 (50) | 14 (11.57) | |
| BMI, kg/m² | 23.28 ± 4.47 | 23.31 ± 4.58 | 23.13 ± 3.90 | 0.8742 | 23.19 ± 4.66 | 23.73 ± 3.37 | 0.6217 | 28.97 ± 1.62 | 23.19 ± 4.44 | 0.0692 |
| Kt/V | 1.77 ± 0.26 | 1.78 ± 0.26 | 1.70 ± 0.23 | 0.2358 | 1.79 ± 0.26 | 1.68 ± 0.25 | 0.0817 | 1.99 ± 0.19 | 1.76 ± 0.26 | 0.2302 |
| ESA | 111 (90.24) | 93 (89.42) | 18 (94.74) | 0.6897 | 92 (89.32) | 19 (95) | 0.6884 | 2 (100) | 109 (90.08) | |
| Oral anticoagulation | 10 (8.13) | 5 (4.81) | 5 (26.32) | 0.0080 | 8 (7.77) | 2 (10) | 0.6654 | 1 (50) | 9 (7.44) | |
| Prior transplant | 4 (3.25) | 4 (3.85) | 0 (0) | | 3 (2.91) | 1 (5) | 0.5130 | 0 (0) | 4 (3.31) | |
| RAASi | 52 (42.28) | 42 (40.38) | 10 (52.63) | 0.3204 | 43 (41.75) | 9 (45) | 0.7876 | 1 (50) | 51 (42.15) | |
| Calcitriol | 55 (44.72) | 45 (43.27) | 10 (52.63) | 0.4504 | 45 (43.69) | 10 (50) | 0.6035 | 2 (100) | 53 (43.8) | |
| Immunosuppressive medication | 7 (5.69) | 6 (5.77) | 1 (5.26) | 0.9302 | 6 (5.83) | 1 (5) | 0.8841 | 0 (0) | 7 (5.79) | |
| Immunodeficiency disorder | 1 (0.81) | 1 (0.96) | 0 (0) | | 1 (0.97) | 0 (0) | | 0 (0) | 1 (0.83) | |
| Albumin, g/dL | 4.17 ± 0.35 | 4.19 ± 0.32 | 4.07 ± 0.49 | 0.3297 | 4.17 ± 0.34 | 4.17 ± 0.42 | 0.9299 | 4.1 ± 0.28 | 4.17 ± 0.35 | 0.7721 |
| Lymphocytes, × 10⁶/L | 1,275 ± 625 | 1,339 ± 647 | 929 ± 317 | <0.0001 | 1,301 ± 640 | 1,145 ± 535 | 0.3084 | 3,086.3 ± 2,510.6 | 1,245.5 ± 537.1 | 0.4884 |
| Hemoglobin, g/dL | 10.70 ± 0.99 | 10.70 ± 1.00 | 10.66 ± 1.01 | 0.8509 | 10.68 ± 0.97 | 10.78 ± 1.14 | 0.7050 | 11.4 ± 0.99 | 10.69 ± 0.99 | 0.3156 |
| BUN, mg/dL | 63.52 ± 14.68 | 63.10 ± 13.31 | 65.79 ± 20.94 | 0.5945 | 63.50 ± 14.06 | 63.60 ± 17.95 | 0.9779 | 69.5 ± 21.92 | 63.42 ± 14.65 | 0.5633 |
| Creatinine, mg/dL | 9.51 ± 2.11 | 9.59 ± 2.02 | 9.07 ± 2.56 | 0.3225 | 9.57 ± 1.96 | 9.19 ± 2.79 | 0.5621 | 9.28 ± 0.67 | 9.51 ± 2.13 | 0.8757 |
| Neutrophils, × 10⁶/L | 3,599 ± 1,412 | 3,523 ± 1,353 | 4,017 ± 1,680 | 0.1613 | 3,574 ± 1,380 | 3,729 ± 1,600 | 0.6539 | 3,115.1 ± 1237.3 | 3,607.1 ± 1417.8 | 0.6270 |
| White blood cells, × 10⁶/L | 5,724 ± 1,791 | 5,713 ± 1,815 | 5,784 ± 1,702 | 0.8,750 | 5,726 ± 1,834 | 5,715 ± 1,596 | 0.9797 | 7,550 ± 4,171.9 | 5,694.2 ± 1,749.5 | 0.6424 |
| Iron (μg/dL) | 71.01 ± 26.42 | 69.83 ± 25.30 | 77.37 ± 31.84 | 0.2551 | 69.34 ± 25.38 | 75.50 ± 30.49 | 0.1163 | 98.5 ± 2.12 | 70.55 ± 26.4 | 0.1385 |
| Ferritin (ng/mL) | 503.87 ± 290.58 | 500.98 ± 305.59 | 519.53 ± 195.20 | 0.7331 | 490.7 ± 282.3 | 570.8 ± 329.1 | 0.2617 | 344.5 ± 143.5 | 506.5 ± 292 | 0.4364 |
| TSAT | 0.33 ± 0.11 | 0.33 ± 0.11 | 0.36 ± 0.15 | 0.3568 | 0.33 ± 0.11 | 0.37 ± 0.13 | 0.1220 | 0.36 ± 0 | 0.33 ± 0.11 | 0.0134 |
| TIBC (μg/dL) | 212.93 ± 36.92 | 212.38 ± 36.98 | 215.95 ± 37.41 | 0.6998 | 212.3 ± 37.54 | 216.10 ± 34.23 | 0.6810 | 273.5 ± 7.78 | 212.5 ± 36 | 0.0186 |
| CRP, mg/L | 6.58 ± 11.05 | 5.90 ± 9.99 | 10.39 ± 15.57 | 0.2518 | 6.02 ± 9.84 | 9.54 ± 16.04 | 0.3660 | 4.95 ± 3.61 | 6.61 ± 11.14 | 0.8347 |
| CRP, median (IQR) | 3.4 (1.8–7.2) | 2.9 (1.8–6.5) | 5.8 (2.1–10.7) | | 3.4 (1.8–6.8) | 3.0 (1.7–7.6) | | 4.95 (2.4–7.5) | 3.4 (1.8–6.8) | |
| Calcium, mg/dL | 9.35 ± 0.51 | 9.35 ± 0.50 | 9.34 ± 0.60 | 0.9569 | 9.34 ± 0.49 | 9.39 ± 0.63 | 0.7250 | 10.25 ± 0.21 | 9.33 ± 0.5 | 0.0113 |
| PTHi, pg/mL | 363.49 ± 424.42 | 373.97 ± 447.31 | 307.76 ± 274.53 | 0.3959 | 374.7 ± 445.1 | 307.2 ± 302.8 | 0.5184 | 784 ± 123 | 356.4 ± 424.3 | 0.1585 |
| PTHi, median (IQR) | 216.50 (128.50–458.00) | 212 (133–466) | 263 (76–446) | | 216.5 (139–458) | 206 (67.3–525) | | 784 (697–871) | 214 (126–447) | |
| nPCR, g/kg/day | 1.09 ± 0.22 | 1.09 ± 0.21 | 1.12 ± 0.28 | 0.6007 | 1.10 ± 0.22 | 1.04 ± 0.21 | 0.2166 | 1.3 ± 0.29 | 1.09 ± 0.22 | |

[a]Values are expressed as means ± the standard deviations or number (%) unless noted otherwise in column 1. BMI, body mass index; BUN, blood urea nitrogen; COPD, chronic pulmonary obstructive disease; CRP, C-reactive protein; ESA, erythropoiesis stimulating agent; IQR, interquartile range; nPCR, normalized protein catabolic rate; PTHi, intact parathyroid hormone; RAASi, renin-angiotensin-aldosterone system inhibitor; sVNT, surrogate viral neutralization test; TIBC, total iron binding capacity; TSAT, transferrin saturation.

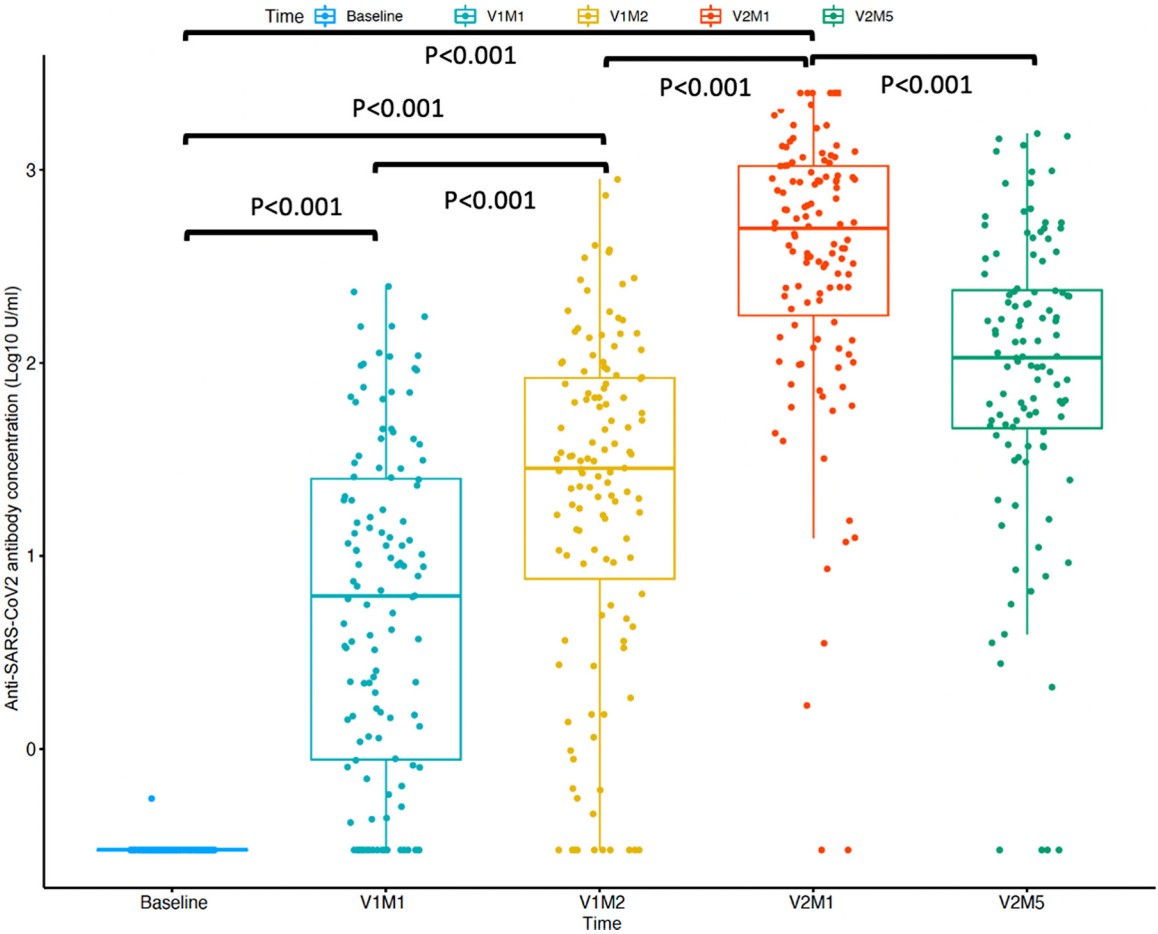

**FIG 1** Trajectories of serum anti-SARS-CoV-2 receptor-binding domain antibodies following two doses of the ChAdOx1 nCoV-19 vaccination in HD patients throughout a 7-month follow-up period. V1M1, 1 month after the first dose of vaccination; V1M2, 2 months after the first dose of vaccination; V2M1, 1 month after the second dose of vaccination; V2M5, 5 months after the second dose of vaccination.

antibody titer significantly decreased to 106.9 U/mL (IQR = 45.3 to 239.5) (Fig. 1), and the seroconversion rate decreased to 96.75%.

**Neutralizing antibodies against ancestral virus and variants of concern.** Neutralizing antibodies against ancestral SARS-CoV-2, delta variant, and omicron variant were detected at V2M1 in 104 (84.6%), 103 (83.7%), and 2 (1.6%) patients, respectively, with a cutoff point of surrogate virus neutralization test (sVNT) inhibition at ≥30% (see Fig. S3). The median cPass readings were 66.5% (IQR = 42.7 to 90.1%), 66.1% (IQR = 41.2 to 83.8%), and 0% (IQR = 0.0 to 9.0%) inhibition against ancestral virus, delta variant, and omicron variant (BA.1), respectively (Fig. 2A). There was no significant difference between the cPass readings for ancestral virus and delta variant, and both were significantly higher than the cPass reading for omicron. At V2M5, the inhibition against ancestral virus and delta variant decreased to 31.2% (IQR = 10.0 to 67.5%) and 8.3% (IQR = 0 to 37.0%), respectively, but neutralization against omicron increased to 15.9% (IQR = 2.6 to 40.5%) (Fig. 2A). Only half of participants (51.8%) remained neutralization against the ancestral virus, and less than one-third of patients were capable of the neutralizing delta and omicron variants (31.6 and 32.5%, respectively).

Furthermore, we randomly selected 40 participants from the study population and measured their serum samples at V2M1 by the pseudovirus microneutralization assay for validation. The 50% pseudovirus microneutralization titers (pVNT50) against the ancestral virus were significantly higher than those against the delta and omicron variants, with the geometric mean titers (GMT) being 639.1, 264.2 and 24.7, respectively (Fig. 2B), equal to 581.5,

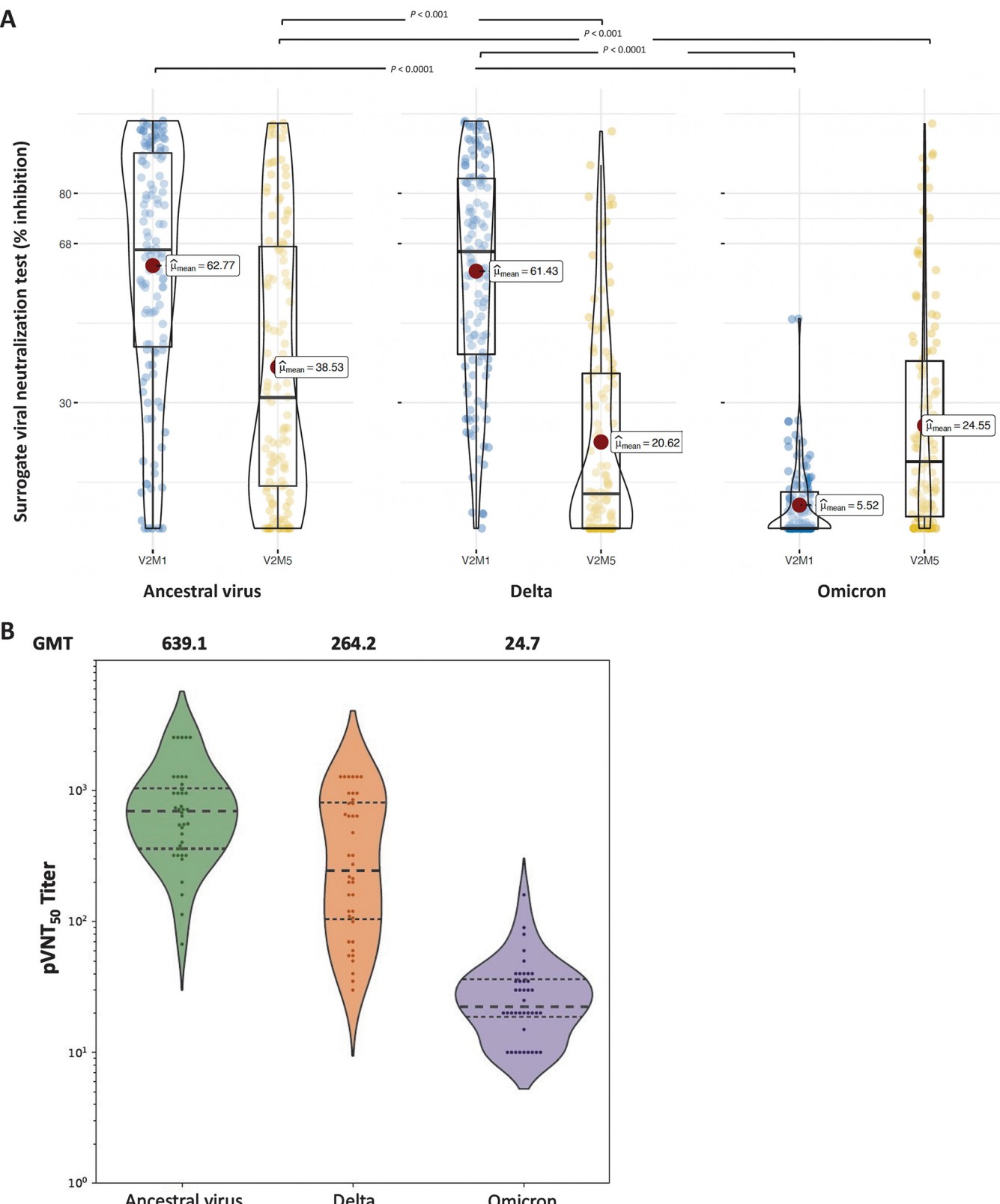

**FIG 2** Neutralization against ancestral SARS-CoV-2 and the delta and omicron variants in HD patients after ChAdOx1 nCoV-19 vaccination. (A) Surrogate viral neutralization test (cPass, inhibition [%]) at 1 and 5 months after the second dose. (B) Pseudovirus neutralization tests (50% pseudovirus neutralization titers [pVNT50]) at 1 month after the second dose of vaccines.

229.6, and 12 IU/mL. The GMT against the ancestral virus was 2.4- and 25.9-fold higher than those against the delta and omicron variants, respectively.

**Correlation between Elecsys anti-SARS-CoV-2 RBD assay and cPass neutralization antibody assay.** The results of the Elecsys assay correlated strongly with the cPass assay in the HD patients, with Spearman rho values of 0.81 and 0.84 for ancestral virus and delta variant sVNTs, respectively (both $P < 0.0001$) (Fig. 3A and B). A cutoff anti-RBD antibody level of 98.6 U/mL had a sensitivity of 0.95 and a specificity of 0.79 to predict a positive sVNT against ancestral virus, and a level of 77.3 U/mL predicted neutralization against delta variant in sVNT with a sensitivity of 0.97 and a specificity of 0.70. However, there was no significant association between the Elecsys assay and omicron sVNT (Spearman rho = $-0.17$, $P = 0.05$) (Fig. 3C).

**Comparison of anti-SARS-CoV-2 RBD antibodies after vaccination between the HD patients and health care workers at V2M5.** A total of 826 of the hospital staff received two doses of AZD1222 and provided blood samples to measure anti-SARS-CoV-2 RBD antibodies at 20 to 24 weeks after vaccination. Since aging is associated with attenuated immunogenicity, we further stratified them by age and compared their antibody data to those of the dialysis patients at V2M5. In those aged younger than 60 years, there was no significant difference between the hospital staff ($n = 749$; median, 174.2 U/mL; IQR = 109.6 to 311.6) and HD patients ($n = 31$; median, 221.4 U/mL; IQR = 85.8 to 410.1) ($P = 0.91$). However, in those older than 60 years, the HD patients had significantly lower titers ($n = 81$; median, 81.7 U/mL; IQR = 33.9 to 202.0) compared to the hospital staff ($n = 77$; median, 138.3 U/mL; IQR = 71.22 to 318.6) ($P = 0.015$) (see Fig. S4).

**Predictors of humoral responses after vaccination.** In the multivariate linear mixed effect model, anti-SARS-CoV-2 RBD antibody titers at V1M1, V1M2, and V2M1 were significantly higher than at baseline, while age and ferritin level were negatively associated with anti-SARS-CoV-2 RBD antibodies. We then used multivariate logistic regression models to investigate the factors predicting the presence or lack of neutralization antibodies. The results showed that low transferrin saturation and C-reactive protein were associated with sVNT inhibition of ≥30% against ancestral virus and delta (Table 2). In addition, higher lymphocyte count and absence of oral anticoagulation or diabetes predicted neutralization against ancestral virus. Older participants were less likely to be elicited neutralizing antibody against delta variant.

We further divided the subjects into two groups according to their anti-SARS-CoV-2 RBD titers at V1M1, and examined the trajectories of antibody levels (see Fig. S5). The results showed that those with V1M1 antibody titers lower than the median, 6.22 U/mL ("low-response group"), had significantly lower antibody titers at each time point, including V1M2, V2M1, and V2M5, and the slopes of the decrease in titers from V2M1 to V2M5 were significantly steeper than in those with V1M1 antibody titers higher than the median ("high-response group") ($P < 0.0001$) (see Table S1 in the supplemental material). In other words, an antibody titer lower than the median at V1M1 predicted lower antibody levels and faster antibody decay during the study period.

## DISCUSSION

To the best of our knowledge, this is the first study to depict the trajectories of anti-SARS-CoV-2 RBD antibodies and investigate their inter-relationships with neutralizing antibodies using sVNT against ancestral virus and the delta and omicron variants after two doses of the adenovector vaccine AZD1222 in infection-naive HD patients. Although nearly all of the HD patients (98.4%) seroconverted, defined as an anti-SARS-CoV-2 RBD titer of ≥0.8 U/mL, the proportion of patients with neutralizing antibodies against ancestral virus or delta or omicron variant in sVNT was lower, and 98.4% of the patients had no neutralization against omicron variant (BA.1). Therefore, our results emphasize that the cutoff value of SARS-CoV-2 RBD titers in HD patients should be set higher than the threshold used in the general population. We suggest that titers of 99 and 77 U/mL are better lower limits for clinical practice to predict protection against ancestral virus and delta variant, respectively, in vaccinated HD patients. Consistent with previous studies, we found an obvious decay in anti-RBD titers with time (a 4.7-fold decrease in 5 months). Currently, 60% of HD

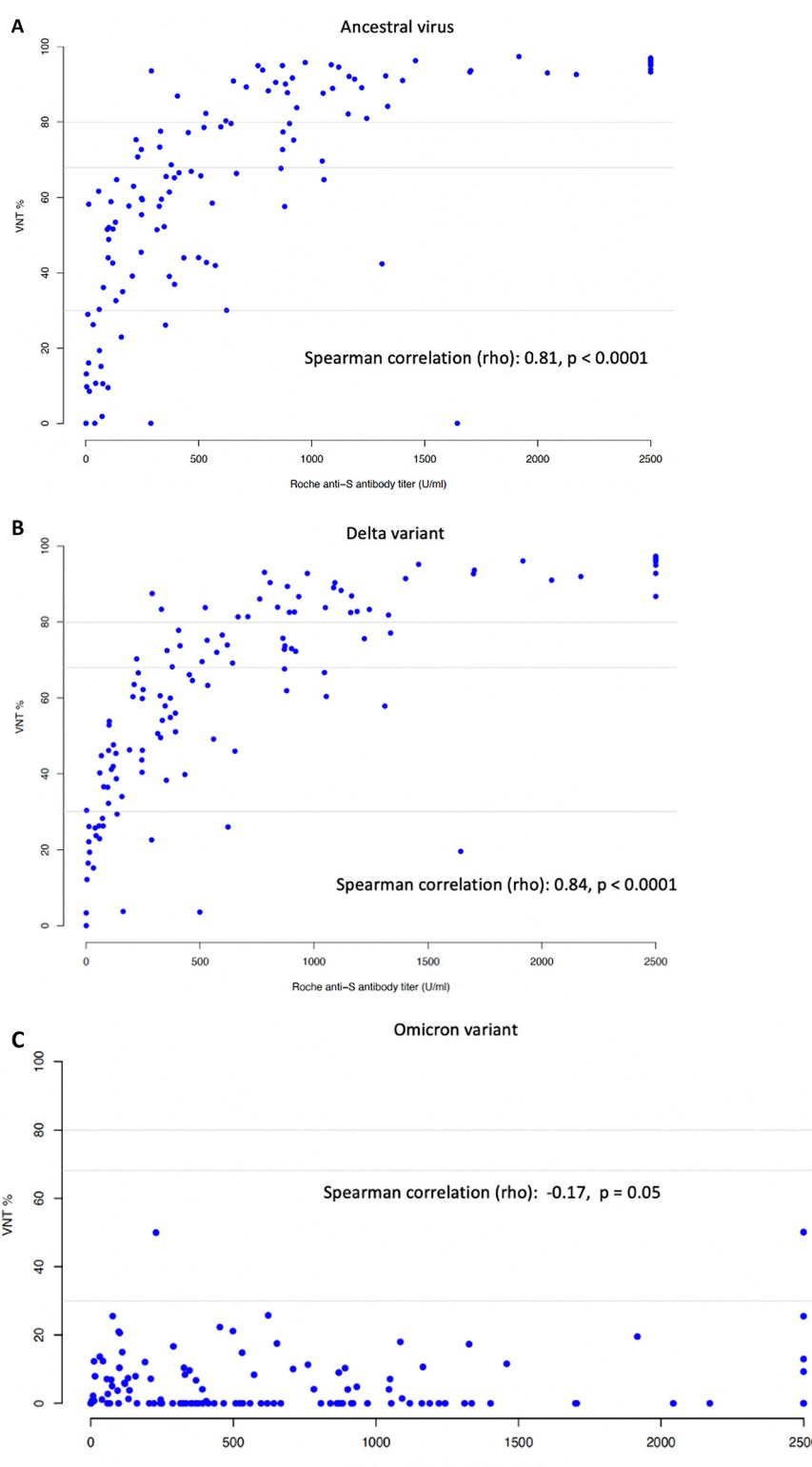

**FIG 3** (A to C) Correlation of anti-SARS-CoV-2 receptor-binding domain antibody levels and neutralizing antibodies (% inhibition) against ancestral SARS-CoV-2 (A), delta variant (B), and omicron variant (C) in HD patients at 1 month after the second dose of ChAdOx1 nCoV-19 vaccine. VNT, viral neutralization test.

**TABLE 2** Estimates for fixed-effect parameters[a]

| Parameter | Antibody titer | | Ancestral virus | | Delta variant | |
|---|---|---|---|---|---|---|
| | Estimate (SE) | P | OR (95% CI) | P | OR (95% CI) | P |
| Time (ref=baseline) | | | | | | |
| V1M1 | 1.25 (0.06) | <0.0001 | | | | |
| V1M2 | 1.82 (0.07) | <0.0001 | | | | |
| V2M1 | 3.06 (0.08) | <0.0001 | | | | |
| Age, yr | −0.01 (0.003) | 0.0044 | | | 0.944 (0.900–0.989) | 0.0158 |
| Lymphocytes | 0.0001 (0.0001) | 0.0879 | 1.002 (1.000–1.004) | 0.0116 | | |
| Ferritin (ng/mL) | −0.0004 (0.0001) | 0.0163 | | | | |
| Oral anticoagulation | | | 0.073 (0.012–0.433) | 0.0039 | | |
| Diabetes mellitus | | | 0.149 (0.031–0.730) | 0.0189 | | |
| TSAT | | | 0.002 (0.001–0.516) | 0.0288 | 0.004 (0.001–0.687) | 0.0354 |
| CRP, mg/L | | | 0.879 (0.796–0.970) | 0.0103 | 0.928 (0.869–0.991) | 0.0260 |

[a]Parameter estimates (and standard errors) for the fixed-effects parameters were obtained by fitting the linear mixed model and predictors for protection against SARS-CoV-2 infection (sVNT inhibition ≥ 30%) by logistic regression models. OR, odds ratio; TAST, transferrin saturation; CRP, C-reactive protein; sVNT, surrogate viral neutralization test; V1M1, 1 month after the first dose of vaccination; V1M2, 2 months after the first dose of vaccination; V2M1, 1 month after the second dose.

patients in the United States and 35% in Taiwan have received no more than two doses of COVID-19 vaccination. Our data highlight the urgent need to enhance protection, such as through additional doses, in these patients, especially as omicron variant is currently the dominant circulating variant globally.

The high seropositive rate of anti-SARS-CoV-2 RBD antibodies following two doses of the AZD1222 vaccination is similar to previous studies of dialysis patients receiving two doses of mRNA vaccines evaluated with anti-RBD or anti-S antibodies (~90% at 1 month) (7). However, a positive anti-RBD or anti-S test does not guarantee neutralization ability in dialysis patients and may overestimate the protection. In two studies of dialysis patients following two doses of mRNA vaccines, neutralization against ancestral virus evaluated using a pseudo-virus neutralization assay and cPass were detected in only 65.4 and 77.8% of the subjects, respectively, despite a high seroconversion rate of anti-S IgG (15, 16). Only 84.6% of the patients in our study had neutralizing antibodies against ancestral virus in sVNT at V2M1. Moreover, almost none of the patients had neutralizing antibodies against omicron, which is consistent with the reported escape of variants of concern (VOCs) from neutralization in previous studies (17–19). However, the neutralization against omicron BA.1 increased 4 months later. It is less likely to be induced by SARS-CoV2 infection because the periodic SARS-CoV-2 PCR or rapid antigen tests during the study period and anti-N antibody level at V2M5 were all negative. Therefore, it may suggest the delayed humoral response in dialysis patients, but more than two-thirds of participants were still not capable of neutralizing omicron. Several studies have demonstrated that additional vaccine doses can increase neutralization against ancestral SARS-CoV-2 and VOCs in dialysis patients (20–23). Carr et al. reported that an additional dose of BNT162b2 following two doses of AZD1222 in HD patients enhanced neutralization, with 67 and 48% of their patients having a 50% inhibitory concentration above 40 against delta and omicron variants, respectively, in live virus microneutralization assays (22). In addition, Cinkilic et al. found that a fourth dose in dialysis patients significantly increased omicron-specific neutralizing antibodies compared to those who received only three doses (23). Another study reported that dialysis patients who had received three doses of mRNA vaccines also had a lower relative risk of omicron infection compared to those who had only received 1 to 2 doses (24). Although the optimal vaccination protocol (dosing, timing, vaccine type, etc.) is unknown, additional doses appears to be a reasonable strategy to reinforce poor neutralization, especially against omicron variant, following two doses of vaccines in dialysis patients.

Although a cutoff value of 0.8 U/mL in the general population has poor specificity for protection in HD patients, the Elecsys SARS-CoV-2-S assay can still be used as a surrogate test to estimate the presence of neutralizing antibodies due to the high correlation with neutralization, as shown in our and other studies (25, 26). In two studies analyzing sera from nondialysis COVID patients, Elecsys titers of 12 and 0.544 U/mL, which are much

lower than in our study, were considered to be optimized cutoff values to predict cPass sVNT $\geq$ 30% (25, 26). The difference may actually be larger, because cPass sVNT probably underestimates neutralization in convalescents compared to vaccinees, since neutralizing antibodies against virus epitopes other than RBD may be induced after natural infection (27). One possible explanation for the higher threshold in our study is decreased avidity or neutralizing capacity of antibodies elicited by vaccination in dialysis patients (15). However, further studies are needed to confirm this hypothesis. In contrast, there was a poor correlation between the Elecsys SARS-CoV-2-S assay and sVNT for omicron variant in our study. Since a correlation between titers of anti-SARS-CoV-2 IgG and neutralizing antibodies against omicron variant has been reported in HD patients receiving three or four doses of vaccines (23), it is possible that the antibody titers in our cohort were too low to demonstrate a correlation.

Our sVNT results revealed similar elicited neutralization capacity against ancestral virus and delta variant, which is in contrast with our pVNT50 data and previous studies. Several studies, including a meta-analysis, reported that either live-virus or pseudovirus neutralization titers against delta variant were at least 3.9- to 4.7-fold lower compared to titers against ancestral virus in either normal or dialysis populations (12, 15, 19). The detection of poorly neutralizing anti-RBD antibodies may lead to false-positive sVNT results (28). In addition, some studies have questioned the correlation between the cPass assay for variants and gold-standard neutralization tests. This assay uses RBD only instead of the full-length S, which may obscure the impact of mutations outside the RBD on the neutralizing ability of the antibodies (29). Further studies are warranted to validate whether sVNT can represent neutralization, especially against VOCs. However, it is still a valuable tool which can provide high-throughput evaluations of neutralization while avoiding the use of live virus and BSL-3 facilities and saving time. In addition, sVNT also facilitates comparability between laboratories (27).

Few studies have investigated immunogenicity after two doses of AZD1222 vaccination or performed comparisons with other vaccines in dialysis patients. In our study, the median anti-SARS-CoV-2 RBD antibody level at V2M1 was 498.80 U/mL, which is between the reported antibody levels following BNT162b2 (25, 171, and 297 U/mL) (30–32) and mRNA-1273 vaccines (1,032 U/mL) (32) in dialysis patients using the same assay as in this study. Using the conversion factor according to the manufacturer's instructions, the binding antibody concentration in our study was 513.2 BAU/mL, while the antibody titers in dialysis patients receiving mRNA vaccines tested with other assays ranged from 39.2 to 1953 BAU/mL (33, 34). With regard to neutralization, one study reported that 77.8% of their HD patients had detectable neutralization against ancestral virus using the cPass assay following two doses of BNT162b2 vaccine, which is lower than in our study (16). In contrast, another study reported that the titers of neutralizing antibodies against delta variant were higher among those who received the BNT162b2 vaccine compared to those who received the AZD1222 vaccine (22). Therefore, it seems that the humoral response elicited by the AZD1222 vaccine may be comparable to mRNA vaccines. However, further studies are needed to confirm this finding due to heterogeneity between previous studies in the study design, immunological assays, and baseline characteristics.

There are several limitations to this study. First, this is a single-center observational study, which may be confounded by selection bias. Therefore, the results are not necessarily generalizable to the whole dialysis population. Second, since no subject in our HD cohort had COVID-19 during the study period, we could not evaluate vaccine effectiveness. However, humoral immunogenicity outcomes have been correlated with the risk of breakthrough infection, hospitalization, and death in previous studies (35–39), and they could be good surrogate parameters of vaccine effectiveness. Third, the control group, our hospital staff members, were younger and had fewer comorbidities than did the HD patients. We stratified both groups by 60 years of age to mitigate the effect of age on immunogenicity. Although some effects may be attributable to

comorbidities instead of kidney failure *per se*, our data represent real-world conditions, since comorbidities such as diabetes and hypertension are prevalent in dialysis patients.

In conclusion, although two doses of the AZD1222 vaccination elicited a seropositive rate of anti-RBD antibodies of 98.4%, around 20% of the patients had no neutralizing antibodies against ancestral virus or delta variant, and only 1.6% of the patients had neutralization against omicron variant in sVNT. Furthermore, anti-RBD antibodies declined substantially with time. Additional doses of vaccines and preventive measures are mandatory to provide adequate protection against COVID-19, especially for emerging VOCs, in dialysis patients receiving two doses of AZD1222.

## MATERIALS AND METHODS

**Study design and vaccination protocol.** In this observational cohort study, the subjects were recruited from the HD center of National Cheng Kung University Hospital (NCKUH), a tertiary medical center in southern Taiwan. Since the incidence of COVID-19 in Taiwan was very low in 2021, the subjects were periodically screened using PCR tests for SARS-CoV-2 to prevent rapid transmission in the HD center. None of the patients at our center had a PCR-confirmed diagnosis of COVID-19 during the study period (June 2021 to January 2022). Since AZD1222 was the most common vaccine provided by the government at that time, all subjects received two doses of AZD1222 vaccination (0.5 mL each, intramuscular injection via the deltoid muscle) with a 2-month interval between doses, between June 2021 and August 2021.

This study was conducted after obtaining the approval of the Institutional Review Board of NCKUH (IRB B-ER-109-024). The inclusion criteria were age older than 20 years and receiving regular HD for more than 3 consecutive months. The exclusion criteria were hospitalization in the previous 3 months and pregnancy. After obtaining informed consent from the participants, we collected blood samples and periodically recorded clinical characteristics and laboratory data. The results of the latest monthly laboratory tests prior to the first dose of the vaccine were defined as the baseline characteristics. Information, including age, sex, dialysis vintage, comorbidities, and medications, were obtained from medical records. Disability was defined as wheelchair use or inability to perform activities of daily living.

**Immunogenicity assessment.** Blood samples were collected at the initiation of the dialysis session for immunogenicity assessment at baseline (within 7 days before the first dose),V1M1 (1 month after the first dose), V1M2 (2 months after the first dose), V2M1 (1 month after the second dose), and V2M5 (5 months after the second dose). We used a US Food and Drug Administration (FDA)-approved electrochemiluminescence immunoassay (Elecsys Anti-SARS-CoV-2 S; Roche Diagnostics, Basel, Switzerland) to detect antibodies to SARS-CoV-2 RBD at these five time points. The analytical measuring interval was 0.4 to 250 U/mL, with a positive threshold of $\geq$0.8 U/mL. Samples with a concentration above the measuring range were diluted 1:10, and the values were reported up to 2,500 U/mL. If the measured antibody level was lower than 0.4 U/mL, it was recorded as 0.3 for the analysis. A US FDA-authorized sVNT assay (cPass SARS-CoV-2 neutralization antibody detection kit; GenScript, Singapore) was used to quantify neutralizing antibodies against ancestral virus and the delta and omicron variants (BA.1) at V2M1 and V2M5. In brief, using purified human angiotensin-converting enzyme 2 (hACE2)-coated enzyme-linked immunosorbent assay plates and horseradish peroxidase-conjugated RBD, the assay detected the presence of immunoglobulins blocking the interaction between hACE2 and RBD. The assay has been shown to be highly correlated with gold-standard live-cell conventional virus and pseudovirus-based neutralization tests (40). With a cutoff at 30% inhibition, cPass was validated in two PCR-confirmed COVID-19 cohorts with 95 to 100% sensitivity and 99.93% specificity (40). From breakthrough infection and outbreak studies in the general population, the manufacturer further defined a cPass reading of >68% inhibition as high and >80% inhibition as very high neutralizing capacity (14, 38, 41).

In addition to sVNT, we also applied pseudovirus microneutralization assay as mentioned previously (42) to V2M1 sera from 40 participants. In brief, the serum samples were first aliquoted and heat-inactivated at 56°C for 30 min. The sera with indicated dilution factors were incubated with 100 tissue culture infectious dose of SARS-CoV-2 spike-expressing pseudovirus for 1 h and then added to 96-well plates of preplated HEK293-ACE O/E cells. After another 18 to 24 h of incubation, the infection rate was evaluated with a luciferase assay system. The pVNT50 titers were derived by curve-fitting functions statistical packages in GraphPad Prism and converted into IU using the convalescent standard sera calibrated by WHO international standard sera (20/130, 20/136, and 20/268) (42).

**Control group.** To compare immunogenicity between the dialysis patients and the general population, we invited the staff at our hospital to undergo tests for anti-RBD antibodies using the Elecsys assay after two doses of the AZD1222 vaccination. Since antibody titers decline after immunization over time, we selected those tested at 20 to 24 weeks after vaccination, and compared them with the titers at V2M5 in our HD cohort. Since aging is associated with attenuated immunogenicity, we further stratified subjects by 60 years old.

**Statistical analysis.** Data were expressed as mean ($\pm$ standard deviation [SD]) or median (IQR) according to parametric or nonparametric distribution, respectively. Results of pVNT50 were illustrated via GMT. Continuous variables were compared between groups using a Student *t* test or a Mann-Whitney U test. Categorical variables were compared using the chi-square test or the Fisher exact test. Dynamic changes in antibody levels throughout the study period were illustrated using a spaghetti plot. Spearman correlation was used to depict the relationship between sVNT and anti-RBD antibodies. Differences in sVNT between

ancestral virus and variants were compared using Tukey *post hoc* analysis on a repeated measures analysis of variance. Independent variables associated with sVNT response were investigated using stepwise multivariate logistic regression models in which the significance level for entry and stay was 0.15. Furthermore, a multivariate linear mixed effect model was constructed to clarify the effect of independent variables on antibody level at various time points because of the nature of repeated measurements. All statistical tests and graphing were performed using SAS version 9.4 (SAS Institute, Cary, NC) or R version 4.1.3. A two-sided *P* value of <0.05 was considered statistically significant.

**Data availability.** The data sets generated and/or analyzed during the present study are not publicly available due to local privacy regulations but are available from the corresponding author on reasonable request.

## SUPPLEMENTAL MATERIAL

Supplemental material is available online only.
**SUPPLEMENTAL FILE 1**, PDF file, 0.9 MB.

## ACKNOWLEDGMENTS

We thank the technical services provided by the Biosafety level 3 core laboratory of the National Core Facility for Biopharmaceuticals, National Science and Technology Council, Taiwan.

We declare that we have no conflicts of interest.

Our research was supported by MOST grants 111-2321-B-006-009-070-MY3 and 111-2314-B-006-070-MY3 from the Ministry of Science and Technology to Y.-T.C.

Research idea and study design (T.-C.L., P.-L.C., N.-Y.L., W.-C.K., and Y.-T.C.), data acquisition (T.-C.L., P.-L.C., C.-Y.S., J.-Y.C., C.-H.C., J.-R.W., and Y.-T.C.), data analysis/interpretation (T.-C.L., P.-L.C., N.-Y.L., W.-C.K., C.-Y.S., J.-Y.C., C.-C.S., C.-F.S., J.-L.W., T.-C.H., C.-H.C., J.-R.W., and Y.-T.C.), statistical analysis (T.-C.L., P.-L.C., N.-Y.L., W.-C.K., J.-L.W., and Y.-T.C.), and supervision or mentorship (Y.-T.C.). All authors contributed important intellectual content during the drafting and revising of the manuscript, accept personal accountability for their own contributions, and agree to ensure that questions pertaining to the accuracy or integrity of any portion of the work have been appropriately investigated and resolved.

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
