## [Reviewer comments · Microbiology Spectrum]

Microbiology Spectrum

Trajectory of humoral responses to two doses of the ChAdOx1 nCoV-19 vaccination in patients receiving maintenance hemodialysis

Tsai-Chieh Ling, Po-Lin Chen, Nan-Yai Li, Wen-Chien Ko, Chien-Yao Sun, Jo-Yen Chao, Chi-Chang Shieh, Ching-Fen Shen, Jia-Ling Wu, Teng-Ching Huang, Chiao-Hsuan Chao, Jen-Ren Wang, and Yu-Tzu Chang

Corresponding Author(s): Yu-Tzu Chang, National Cheng Kung University Hospital

Review Timeline:

Submission Date:	September 2, 2022
Editorial Decision:	November 28, 2022
Revision Received:	January 26, 2023
Accepted:	January 27, 2023

Editor: Leonidas Stamatatos

Reviewer(s): Disclosure of reviewer identity is with reference to reviewer comments included in decision letter(s). The following individuals involved in review of your submission have agreed to reveal their identity: Luciana Jesus Costa (Reviewer #1); Jianjun Chen (Reviewer #2)

Transaction Report:

DOI: <https://doi.org/10.1128/spectrum.03445-22>

November 28, 2022

Dr. Yu-Tzu Chang
National Cheng Kung University Hospital and Medical College
Department of Internal Medicine
Tainan
Taiwan

Re: Spectrum03445-22 (Trajectory of humoral responses to two doses of the ChAdOx1 nCoV-19 vaccination in patients receiving maintenance hemodialysis)

Dear Dr. Yu-Tzu Chang:

Link Not Available

Sincerely,

Leonidas Stamatatos

Journals Department
Reviewer comments:

Reviewer #1 (Comments for the Author):

The manuscript by Ling and co-workers is relevant in the context of the magnitude of vaccine response in specific subpopulations. The data is sound, and several important comparisons are presented. A missing data that would increase the relevance of the study would be the sVNT from V2M5 samples. The waning of anti-RBD levels from V2M1 to V2M5 was about $1/2 \log_{10}$, which predicts protection levels still, from what was mentioned from authors. Thus, knowing the levels of neutralizing antibodies in these samples will help to understand the dynamics of protection in this population.

On the sentence "Therefore, our results emphasize that the cut-off value of SARS-CoV-2 RBD titers in HD patients should be

set higher than the threshold used in the general population."(lanes 187-188), the relationship is not clear, please explain. If the authors are referring to the correlation found between levels of anti-RBD antibodies and neutralizing antibodies (by sVNT), there was a large range of antibody levels, varying from ~750-2500 U/mL, with 85-100% neutralization. Thus, which should be the factor to set threshold higher for the HD patients?

Paper quality would be improved if results section is restructured, for instance, adverse effects of vaccination should come together with baseline characteristics description.

In baseline characteristics authors already introduced the relationship found between neutralization levels and age and diabetes status, without having introduced the presence of antibody levels and neutralization antibody presence (Lanes 108-109) and did not explore these correlations further.

Minor:

1) The cited reference and references within refer to COVID-19 frequency, hospitalization and mortality rates amongst kidney failure patients in specific populations (US and Europe), the way it is phrased induces an understanding of global rates, thus authors should rephrase it to specify the specificity of the data mentioned or provide other references indicating the same frequencies in global populations.

Reviewer #2 (Comments for the Author):

This work describes the humoral responses in dialysis patients after two doses of AZD122 vaccination. Patients with kidney failure have worse immune response compared to healthy person. Many of them are elderly with multiple comorbidities. These made them more prone to SARS-CoV-2 infection. This work demonstrated that two dose of AZD1222 vaccines lead to high titer of antibodies against prototype and delta RBD, but low titer of antibody against Omicron RBD in dialysis patients with multiple comorbidities. Also, this work showed that there was a substantial decay in anti-RBD titers with time. This manuscript provided some useful and interesting clinical data of dialysis patients. However, some specific comments are listed as follows:

- 1、 Lane 91, the studies of mRNA vaccine in dialysis person should be cited.
- 2、 SARS-CoV-2 live virus neutralization assay or at least pseudovirus neutralization assay should be needed.

Staff Comments:

Preparing Revision Guidelines

Please return the manuscript within 60 days; if you cannot complete the modification within this time period, please contact me. If you do not wish to modify the manuscript and prefer to submit it to another journal, please notify me of your decision immediately so that the manuscript may be formally withdrawn from consideration by Microbiology Spectrum.

The manuscript by Ling and co-workers is relevant in the context of the magnitude of vaccine response in specific subpopulations. The data is sound, and several important comparisons are presented. A missing data that would increase the relevance of the study would be the sVNT from V2M5 samples. The waning of anti-RBD levels from V2M1 to V2M5 was about $\frac{1}{2} \log_{10}$, which predicts protection levels still, from what was mentioned from authors. Thus, knowing the levels of neutralizing antibodies in these samples will help to understand the dynamics of protection in this population.

On the sentence “Therefore, our results emphasize that the cut-off value of SARS-CoV-2 RBD titers in HD patients should be set higher than the threshold used in the general population.”(lanes 187-188), the relationship is not clear, please explain. If the authors are referring to the correlation found between levels of anti-RBD antibodies and neutralizing antibodies (by sVNT), there was a large range of antibody levels, varying from ~ 750 -2500 U/mL, with 85-100% neutralization. Thus, which should be the factor to set threshold higher for the HD patients?

Paper quality would be improved if results section is restructured, for instance, adverse effects of vaccination should come together with baseline characteristics description.

In baseline characteristics authors already introduced the relationship found between neutralization levels and age and diabetes status, without having introduced the presence of antibody levels and neutralization antibody presence (Lanes 108-109) and did not explore these correlations further.

Minor:

- 1) The cited reference and references within refer to COVID-19 frequency, hospitalization and mortality rates amongst kidney failure patients in specific populations (US and Europe), the way it is phrased induces an understanding of global rates, thus authors should rephrase it to specify the specificity of the data mentioned or provide other references indicating the same frequencies in global populations.

Microbiology Spectrum Editorial Team,

Re: Spectrum03445-22

My colleagues and I appreciate very much the constructive review comments on our manuscript by the editors and the two reviewers, which actually guide the revision. Without their useful advices, the manuscript would not have been improved so much. We have made several major changes by following their suggestions and attached below. Please find the point-by-point responses to each of the comments from the reviewers. The alterations made in the revised manuscript are marked in red and are also listed after each response. Thanks again for your time and consideration. If more revision is needed, please kindly let us know.

Sincerely yours,

Yu-Tzu Chang, M.D., Ph.D.,

Department of Internal Medicine, National Cheng Kung University Hospital, College of Medicine, National Cheng Kung University, 138 Shang-Li Rd., Tainan 70428, Taiwan.

Reviewer comments:

Reviewer #1 (Comments for the Author):

1. The manuscript by Ling and co-workers is relevant in the context of the magnitude of vaccine response in specific subpopulations. The data is sound, and several important comparisons are presented. A missing data that would increase the relevance of the study would be the sVNT from V2M5 samples. The waning of anti-RBD levels from V2M1 to V2M5 was about 1/2 log₁₀, which predicts protection levels still, from what was mentioned from authors. Thus, knowing the levels of neutralizing antibodies in these samples will help to understand the dynamics of protection in this population.

Response: Thanks for your suggestion. Since we've known the anti-RBD antibody titers waned with time after vaccination (Figure 1), it's important to figure out the dynamic change of the neutralizing antibodies concentration, especially at the time point of V2M5. By following your suggestion, we checked the sVNT at V2M5. The median cPass reading were 31.2, 8.3 and 15.9% inhibition against the ancestral virus, delta and omicron (BA.1), respectively. The neutralizing antibodies against the ancestral virus were significantly higher than those against delta or omicron BA.1 at V2M5. The neutralizing antibodies against ancestral virus and delta decreased with time during V2M1 to V2M5. However, surprisingly, the level of sVNT against omicron BA.1 increased from V2M1 to V2M5. The participants in our study have received SARS-CoV-2 PCR or rapid antigen tests periodically during the study period, and the anti-N antibody of these patients at V2M5 were all negative. Thus, the increase of neutralization capacity against omicron was not likely to be induced by natural infection. It has been suggested that delayed humoral response elicited by COVID-19 vaccination existed in hemodialysis patients as compared with those of general population.(1-3) Further studies are warranted to evaluate when the neutralizing antibodies against variants peak after vaccination in dialysis patients. To facilitate the comparison of antibody level between ours and others, including the data derived from dialysis patients or general population, the neutralizing antibody levels at one month after vaccination are still valuable. By following your suggestion, we've modified several parts of the manuscript as follows:

Results (Page 7, Line 129)

Neutralizing antibodies against ancestral virus and the variants of concern (VOCs)

Neutralizing antibodies against ancestral SARS-CoV-2, delta and omicron were detected at V2M1 in 104 (84.6%), 103 (83.7%) and 2 (1.6%) patients, respectively, with a cut-off point of surrogate virus neutralization test (sVNT) inhibition $\geq 30\%$ (Supplementary Figure 3). The median cPass readings were 66.5% (IQR: 42.7-90.1%), 66.1% (IQR: 41.2-83.8%), and 0% (IQR: 0.0-9.0%) inhibition against ancestral virus, delta and omicron (BA.1), respectively (Figure 2A). There was no significant difference between the cPass readings for ancestral virus and delta, and both were significantly higher than the cPass reading for omicron. At V2M5, the inhibition against ancestral virus and delta decreased to 31.2 % (IQR: 10.0-67.5%) and 8.3 % (IQR: 0-37.0%), respectively, but neutralization against omicron increased to 15.9 % (IQR: 2.6-40.5%). (Figure 2A). Only half of participants (51.8%) remained neutralization against the ancestral virus, and less than one-third of patients were capable of neutralizing delta and omicron (31.6 and 32.5%, respectively).

Discussion (Page 10, Line 209)

The high seropositive rate of anti-SARS-CoV-2 RBD antibodies following two doses of the AZD1222 vaccination is similar to previous studies of dialysis patients receiving two doses of mRNA vaccines evaluated with anti-RBD or anti-S antibodies (~90% at 1 month).(7) However, a positive anti-RBD or anti-S test does not guarantee neutralization ability in dialysis patients, and may overestimate the protection. In two studies of dialysis patients following two doses of mRNA vaccines, neutralization against ancestral virus evaluated using a pseudovirus neutralization assay and cPass were detected in only 65.4% and 77.8% of the subjects, respectively, despite a high seroconversion rate of anti-S IgG.(16, 17) Only 84.6% of the patients in our study had neutralizing antibodies against ancestral virus in sVNT at V2M1. Moreover, almost none of the patients had neutralizing antibodies against omicron, which is consistent with the reported escape of VOCs from neutralization in previous studies.(18-20) However, the neutralization against omicron BA.1 increased 4 months later. It is less likely to be induced by SARS-CoV2 infection because the periodic SARS-CoV-2 PCR or rapid antigen tests during the study period and anti-N antibody level at V2M5 were all negative. Therefore, it may suggest the delayed humoral response in dialysis patients, but more than two-thirds of participants were still not capable of neutralizing omicron.

Method (Page 15, Line 328)

Immunogenicity assessment

Blood samples were collected at the initiation of the dialysis session for immunogenicity assessment at baseline (within 7 days before the first dose), V1M1 (1 month after the first dose), V1M2 (2 months after the first dose), V2M1 (1 month after the second dose), and V2M5 (5 months after the second dose). We used a US-FDA-approved electro-chemiluminescence immunoassay (Elecsys® Anti-SARS-CoV-2 S; Roche Diagnostics, Basel, Switzerland) to detect antibodies to SARS-CoV-2 RBD at these five time points. The analytical measuring interval was 0.4 U/ml to 250 U/ml, with a positive threshold of ≥ 0.8 U/ml. Samples with a concentration above the measuring range were diluted 1:10, and the values were reported up to 2500 U/ml. If the measured antibody level was lower than 0.4 U/ml, it was recorded as 0.3 for the analysis. A US-FDA-authorized sVNT assay (cPass SARS-CoV-2 Neutralization Antibody Detection Kit, GenScript, Singapore) was used to quantify neutralizing antibodies against ancestral virus, delta and omicron (BA.1) at V2M1 and V2M5.

Figure 2. Neutralization against ancestral SARS-CoV-2, delta and omicron variants in hemodialysis patients after ChAdOx1 nCoV-19 vaccination: (2A) Surrogate viral neutralization test (cPass, inhibition %) at one and five months after the second dose and (2B) pseudovirus neutralization tests (50% pseudovirus neutralization titers [pVNT50]) at 1 month after the second dose of vaccines.

Figure 2A.

- On the sentence "Therefore, our results emphasize that the cut-off value of SARS-CoV-2 RBD titers in HD patients should be set higher than the threshold used in the general population." (lanes 187-188), the relationship is not clear, please explain. If the authors are referring to the correlation found between levels of anti-RBD antibodies and neutralizing antibodies (by sVNT), there was a large range of antibody levels, varying from ~750-2500 U/mL, with 85-100% neutralization. Thus, which should be the factor to set threshold higher for the HD patients?

Response: Thank you for your question. According to several study results derived from the general population, the optimal cutoffs of anti-RBD antibody titer to predict the presence of neutralizing antibodies (cPass sVNT \geq 30%) could be as low as 0.544 U/ml. (Lanes 223-226) However, our data revealed that in our dialysis cohort, the optimal threshold to predict neutralization against ancestral virus was 98.6 U/ml, with a sensitivity of 0.95 and specificity of 0.79. (Lanes 137-140) Therefore, it is the reason why we suggested the threshold of anti-RBD antibodies for protection should be set higher in patients on maintenance hemodialysis, instead of 0.8 U/ml applied in the general population suggested by manufacturer, to avoid overestimate protection.

- Paper quality would be improved if results section is restructured, for instance, adverse effects of vaccination should come together with baseline characteristics description.

Response: Thank you for your suggestion. By following your suggestion, we merged the adverse effect section into the first paragraph of results, and the manuscript is modified as follows:

Results (Page 6, Line 105)

Baseline characteristics and adverse events following vaccination

A total of 131 HD patients were initially enrolled, of whom 123 completed the scheduled blood sampling protocol (Supplement Figure 1). The baseline characteristics are shown in Table 1. There were significant differences in age and diabetes between the groups with inhibition $\geq 30\%$ and $< 30\%$ against the ancestral virus and delta variant.

The most common adverse reaction to the vaccine following the first dose was fever, which developed in 26 (21.1%) patients, followed by headache (4.9%), injection site pain (3.3%), fatigue (3.3%) and arthralgia/myalgia (3.3%) (Supplement Figure 2). Following the second dose, only two (1.6%) patients developed fever. No serious adverse events were noted during the study period.

- 4. In baseline characteristics authors already introduced the relationship found between neutralization levels and age and diabetes status, without having introduced the presence of antibody levels and neutralization antibody presence (Lanes 108-109) and did not explore these correlations further.**

Response: Thank you for your comments. We have found that age and percentage of diabetes mellitus at baseline in patients with neutralizing antibodies against ancestral or delta virus (sVNT cPass reading $\geq 30\%$) at V2M1 were significantly lower than those without neutralizing antibodies. (Table 1) However, when selecting independent variables by using the stepwise multivariate logistic regression models, in which the significance level for entry and stay was 0.15, age and diabetes were not universally selected into the analysis of predictors for neutralization against ancestral and delta virus, respectively. In the final results of the regression models, independent variables with statistical significance to predict neutralization against ancestral virus were higher lymphocyte count, lower transferrin saturation (TSAT), lower CRP and absence of anticoagulant use or diabetes. On the other hand, younger age, lower TSAT and lower CRP were associated with neutralization against delta. To make it clear, we modified the manuscript as follows:

Results (Page 8, Line 171)

Predictors of humoral responses after vaccination

In the multivariate linear mixed effect model, anti-SARS-CoV-2 RBD antibody titers at V1M1, V1M2 and V2M1 were significantly higher than at baseline, while age and ferritin level were negatively associated with anti-SARS-CoV-2 RBD antibodies. We then used multivariate logistic regression models to investigate the factors predicting the presence or lack of neutralization antibodies. The results showed that low transferrin saturation and C-reactive protein were associated with sVNT inhibition $\geq 30\%$ against ancestral virus and delta (Table 2). **In addition, higher lymphocyte count and absence of oral anticoagulation or diabetes predicted neutralization against ancestral virus. Older participants were less likely to be elicited neutralizing antibody against delta.**

Minor:

- 1. The cited reference and references within refer to COVID-19 frequency, hospitalization and mortality rates amongst kidney failure patients in specific populations (US and Europe), the way it is phrased induces an understanding of global rates, thus authors should rephrase it to specify the specificity of the data mentioned or provide other references indicating the same frequencies in global populations.**

Response: Thanks for your suggestion, and we totally agree with it. To provide a global picture to readers, we added several references regarding data from places other than US and Europe. According to a web-survey of ISN and Dialysis Outcomes Practice Patterns Study, morality of in-center hemodialysis patients was reported to be $\geq 10\%$ in most centers, but in Latin America and Africa, $\sim 1/4$ HD centers reported mortality rates $> 50\%$.(4) We've modified the manuscript as follows:

Introduction (Page 4, Line 76)

The coronavirus disease 2019 (COVID-19) pandemic poses a major threat to patients with chronic kidney disease, as they are more prone to SARS-CoV-2 infection and the subsequent complications, including hospitalization, respiratory failure and mortality. Patients treated with in-center hemodialysis (HD) are particularly vulnerable to COVID-19, since they cannot self-isolate during the dialysis procedure, and many of them are elderly with multiple comorbidities.(1) An estimated 28-36% of kidney failure patients were infected during the first wave of the pandemic, which is 5 to 20 times higher than in the general population(2), and the mortality rate was as high as 20-30% in Europe (3), or even higher than 50% at some regions with lower income (4).

References (Page 20, Line 417)

4. Aylward R, Bieber B, Guedes M, Pisoni R, Tannor EK, Dreyer G, Liew A, Luyckx V, Shah DS, Phiri C, Evans R, Albakr R, Perl J, Jha V, Pecoits-Filho R, Robinson B, Caskey FJ. 2022. The Global Impact of the COVID-19 Pandemic on In-Center Hemodialysis Services: An ISN-Dialysis Outcomes Practice Patterns Study Survey. *Kidney Int Rep* 7:397-409.

Reviewer #2 (Comments for the Author):

This work describes the humoral responses in dialysis patients after two doses of AZD122 vaccination. Patients with kidney failure have worse immune response compared to healthy person. Many of them are elderly with multiple comorbidities. These made them more prone to SARS-CoV-2 infection. This work demonstrated that two dose of AZD1222 vaccines lead to high titer of antibodies against prototype and delta RBD, but low titer of antibody against Omicron RBD in dialysis patients with multiple comorbidities. Also, this work showed that there was a substantial decay in anti-RBD titers with time. This manuscript provided some useful and interesting clinical data of dialysis patients. However, some specific comments are listed as follows:

1. Lane 91, the studies of mRNA vaccine in dialysis person should be cited.

Response: Thanks for your suggestion, and we totally agree with that. The manuscript is modified as follows:

Introduction (Page 4, Line 85)

Several SARS-CoV-2 vaccines have been proven to reduce the risks of infection and severe disease in the general population, however none of the efficacy trials included dialysis patients. Since responses to vaccines such as hepatitis B and influenza(4, 5) are weaker in patients undergoing dialysis, it is important to investigate whether authorized SARS-CoV-2 vaccines provide adequate protection in this population. Most studies investigating immunogenicity in dialysis patients have focused on humoral responses to structural proteins (i.e., spike protein [S] or receptor-binding domain [RBD]) following mRNA vaccines(7, 8).

References (Page 21, Line 428)

7. Chen JJ, Lee TH, Tian YC, Lee CC, Fan PC, Chang CH. 2021. Immunogenicity Rates After SARS-CoV-2 Vaccination in People With End-stage Kidney Disease: A Systematic Review and Meta-analysis. *JAMA Netw Open* 4:e2131749.

8. Ma BM, Tam AR, Chan KW, Ma MKM, Hung IFN, Yap DYH, Chan TM. 2022. Immunogenicity and Safety of COVID-19 Vaccines in Patients Receiving Renal Replacement Therapy: A Systematic Review and Meta-Analysis. *Front Med (Lausanne)* 9:827859.

2. SARS-CoV-2 live virus neutralization assay or at least pseudovirus neutralization assay should be needed.

Response: Thanks for your constructive suggestion, which makes our work more comprehensive. Although the surrogate neutralization test has been proved to be correlated with live-cell conventional virus and pseudovirus-based neutralization tests, the gold standard neutralization tests are the latter. To evaluate the neutralization capacity by the pseudovirus-based neutralization test at the time point of V2M1, we randomly selected 40 participants from the study population. Our results revealed that the 50% pseudovirus neutralization titers (pVNT50) against the ancestral virus were significantly higher than those against delta and omicron BA.1, with the geometric mean titer (GMT) being 639.1, 264.2 and 24.7, respectively. In previous literatures from general population, 8 and 54 international unit (IU)/ml could be the cut-off values of neutralizing antibody levels for protection from severe infection and symptomatic infection, respectively.(5) After conversion of our study results to IU/ml, it's estimated only 7.5% participants gained protection from symptomatic infection of omicron. On the other hand, 100% and 82.5% of them were protected from symptomatic infection of the ancestral virus and delta, respectively. In perspective of protection against severe infection, 100%, 100% and 75% of them gained corresponding neutralizing titers against the ancestral virus, delta and omicron, respectively. However, this is only a rough estimate, since dialysis and elderly people were not included in the data sources for modeling relationships between disease outcomes and titers, and immunological mechanisms other than humoral immunity also play important roles in providing protection. Our pVNT50 data underlines the escape of variants from neutralization, and the conversion into international unit facilitate the comparisons with other studies. By following your suggestion, we've modified several parts of the manuscript as follows:

Abstract (Page 2, Line 48)

At 1 month after the second dose, 84.6%, 83.7%, and 1.6% of the participants had neutralizing antibodies against the ancestral virus, delta, and omicron, respectively, measured by a commercial surrogate neutralization assay. **The geometric mean of 50% pseudovirus neutralization titers for the ancestral virus, delta, and omicron were 639.1, 264.2 and 24.7, respectively.**

Importance (Page 3, Line 63)

Patients with kidney failure have worse immune response following vaccination compared to general population, but few clinical studies have investigated immunogenicity of ChAdOx1 nCoV-19 (AZD1222) vaccination in hemodialysis patients. Here, we showed two doses of AZD1222 vaccines lead to high seroconversion rate of anti-SARS-CoV-2 receptor binding

domain (RBD) antibodies, and more than 80% patients acquired neutralizing antibodies against ancestral virus and delta variant. However, seldom did they obtain neutralizing antibodies against the omicron variant. **The geometric mean of 50% pseudovirus neutralization titers against the ancestral virus was 25.9-fold higher than that against omicron.** Also, there was a substantial decay in anti-RBD titers with time. Our findings provided evidence supporting that more protective measures, included additional/booster vaccinations, is warranted in these patients during the current COVID-19 pandemic.

Results (Page 7, Line 142)

Furthermore, we randomly selected 40 participants from the study population and measured their serum samples at V2M1 by the pseudovirus micro-neutralization assay for validation. The 50% pseudovirus micro-neutralization titers (pVNT50) against the ancestral virus were significantly higher than those against delta and omicron, with the geometric mean titer (GMT) being 639.1, 264.2 and 24.7, respectively (Figure 2B), equal to 581.5, 229.6 and 12 IU/ml. The GMT against the ancestral virus was 2.4- and 25.9-fold higher than those against delta and omicron, respectively.

Discussion (Page 12, Line 254)

Our **sVNT** results revealed similar elicited neutralization capacity against ancestral virus and delta, which is in contrast with **our pVNT50 data and** previous studies. Several studies, including a meta-analysis, reported that either live-virus or pseudovirus neutralization titers against delta were at least 3.9-4.7 fold lower compared with titers against ancestral virus in either normal or dialysis populations.(12, 16, 20)

Discussion (Page 13, Line 285)

There are several limitations to this study. First, this is a single-center observational study which may be confounded by selection bias. Therefore, the results are not necessarily generalizable to the whole dialysis population. Second, since no subject in our HD cohort had COVID-19 during the study period, we could not evaluate vaccine effectiveness. However, humoral immunogenicity outcomes have been correlated with the risk of breakthrough infection, hospitalization and death in previous studies,(36-40) and they could be good surrogate parameters of vaccine effectiveness. **Third**, the control group, our hospital members of staff, were younger and had fewer comorbidities than the HD patients. We stratified both groups by 60 years of age to mitigate the effect of age on immunogenicity. Although some effects may be attributable to comorbidities instead of kidney failure per se, our data represent real-world conditions, since comorbidities such as diabetes and hypertension are prevalent in dialysis patients.

Discussion (Page 14, Line 297)

In conclusion, although two doses of the AZD1222 vaccination elicited a seropositive rate of anti-RBD antibodies of 98.4%, around 20% of the patients had no neutralizing antibodies against ancestral virus or delta variant, and only 1.6% of the patients had neutralization against omicron **in sVNT**. Furthermore, anti-RBD antibodies declined substantially with time. Additional doses of vaccines and preventive measures are mandatory to provide adequate protection against COVID-19, especially for emerging VOCs, in dialysis patients receiving two doses of AZD1222.

Method (Page 16, Line 351)

In addition to sVNT, we also applied the pseudovirus micro-neutralization assay as mentioned previously(43) to V2M1 sera from forty participants. In brief, the serum samples were first aliquoted and heat-inactivated at 56°C for 30 min. The sera with indicated dilution factors were incubated with 100 tissue culture infectious dose of SARS-CoV-2 spike-expressing pseudovirus for one hour, and then added to 96-well plates of pre-plated HEK293-ACE O/E cells. After another 18-24 h of incubation, the infection rate was evaluated with a luciferase assay system. The pVNT50 titers were derived by curve-fitting functions statistical packages in GraphPad Prism, and converted into IU using the convalescent standard sera calibrated by WHO international standard sera (20/130, 20/136, and 20/268).(43)

Method (Page 17, Line 369)

Statistical analysis

Data were expressed as mean (\pm standard deviation [SD]) or median (interquartile range [IQR]) according to parametric or non-parametric distribution, respectively. Results of pVNT50 were illustrated via GMT. Continuous variables were compared between groups using the Student's t-test or Mann–Whitney U test. Categorical variables were compared using the chi-square test or Fisher's exact test.

References (Page 29, Line 589)

43. Chao CH, Cheng D, Huang SW, Chuang YC, Yeh TM, Wang JR. 2022. Serological responses triggered by different SARS-CoV-2 vaccines against SARS-CoV-2 variants in Taiwan. *Front Immunol* 13:1023943.

Figure 2. Neutralization against ancestral SARS-CoV-2, delta and omicron variants in hemodialysis patients after ChAdOx1 nCoV-19 vaccination: (2A) Surrogate viral neutralization test (cPass, inhibition %) at one and five months after the second dose and (2B) pseudovirus neutralization tests (50% pseudovirus neutralization titers [pVNT50]) at 1 month after the second dose of vaccines.

Figure 2B.

References:

1. Labriola L, Scohy A, Van Regemorter E, Robert A, Clerbaux G, Gillerot G, Pochet JM, Biller P, De Schuiteneer M, Morelle J, Yombi JC, Kabamba B, Rodriguez-Villalobos H, Jadoul M. 2021. Immunogenicity of BNT162b2 SARS-CoV-2 Vaccine in a Multicenter Cohort of Nursing Home Residents Receiving Maintenance Hemodialysis. *Am J Kidney Dis* 78:766-768.
2. Simon B, Rubey H, Treipl A, Gromann M, Hemedi B, Zehetmayer S, Kirsch B. 2021. Haemodialysis patients show a highly diminished antibody response after COVID-19 mRNA vaccination compared with healthy controls. *Nephrol Dial Transplant* 36:1709-1716.
3. Ikizler TA, Coates PT, Rovin BH, Ronco P. 2021. Immune response to SARS-CoV-2 infection and vaccination in patients receiving kidney replacement therapy. *Kidney Int* 99:1275-1279.
4. Aylward R, Bieber B, Guedes M, Pisoni R, Tannor EK, Dreyer G, Liew A, Luyckx V, Shah DS, Phiri C, Evans R, Albakr R, Perl J, Jha V, Pecoits-Filho R, Robinson B, Caskey FJ. 2022. The Global Impact of the COVID-19 Pandemic on In-Center Hemodialysis Services: An ISN-Dialysis Outcomes Practice Patterns Study Survey. *Kidney Int Rep* 7:397-409.
5. Khoury DS, Cromer D, Reynaldi A, Schlub TE, Wheatley AK, Juno JA, Subbarao K, Kent SJ, Triccas JA, Davenport MP. 2021. Neutralizing antibody levels are highly predictive of immune protection from symptomatic SARS-CoV-2 infection. *Nat Med* 27:1205-1211.

January 27, 2023

Dr. Yu-Tzu Chang
National Cheng Kung University Hospital
Department of Internal Medicine
Tainan
Taiwan

Re: Spectrum03445-22R1 (Trajectory of humoral responses to two doses of the ChAdOx1 nCoV-19 vaccination in patients receiving maintenance hemodialysis)

Dear Dr. Yu-Tzu Chang:

You and your colleagues carefully addressed the comments made by the reviewer and appropriately edited their manuscript. As a result, your revised manuscript vastly improved.

Your manuscript has been accepted, and I am forwarding it to the ASM Journals Department for publication. You will be notified when your proofs are ready to be viewed.

Sincerely,

Leonidas Stamatatos
Editor, Microbiology Spectrum
